# Evaluation for the Leaching of Cr from Coal Gangue Using Expansive Soils

**Yan Zhang [1,2,\*], Hassan Baaj [2] and Rong Zhao [1]**

[1]  College of Energy and Transportation Engineering, Inner Mongolia Agricultural University, Hohhot 010018, China

[2]  Department of Civil & Environmental Engineering, University of Waterloo, Waterloo, ON N2L 3G1, Canada

\*  Correspondence: zhangyanli@imau.edu.cn; Tel.:+1-0-471-530-7514

**Abstract:** Coal gangue can cause significant heavy metal pollution in mining areas, which would have a negative impact on the environment and human health. The objective of this research is to investigate the relationship between expansive soil amount and the leaching behavior of Chromium from coal gangue and the engineering properties of coal gangue used as building materials. The leaching behavior of Chromium from coal gangue was observed using atomic absorption spectrometry. A column leaching experiment was conducted to examine the impact of leaching time and heavy metal concentration. Furthermore, the unconfined compressive strength test was employed to evaluate the engineering properties of coal gangue with expansive soil. The results of the study demonstrate that pH of leachate solutions, leaching time, and expansive soil amounts in mixtures have important influence on Chromium concentration. The leachate solutions, which behave like alkaline, provide a positive environment for adsorbing Cr. Adding expansive soil can reduce leached concentrations of Chromium from coal gangue when compared to leachate of original coal gangue. It was found that 30% expansive soil was an improved solution because it delayed the cumulative concentration to reach the limitation line. Moreover, the unconfined compressive strength of coal gangue was boosted through adding expansive soil.

**Keywords:** coal gangue; expansive soil; leaching; heavy metal; unconfined compressive strength

## 1. Introduction

Coal gangue (CG), the main waste of the coal mining industry, accounts for about 10–15% of raw coal mined. In 2016, China, the largest coal producer in the world, mined about 310 Mt of coal, resulting in 31–46.5 Mt CG. A huge amount of this CG was stacked to form solid waste dumps, which occupied a great deal of land and was potentially hazardous [1–4]. Such coal mining waste has the potential to pollute the air, the soil, and the underground water through self-ignition and leaching processes and could consequently affect the environment of the biosphere [5,6]. Some harmful heavy metals, such as chromium, arsenic, mercury, lead, and cadmium [7–10], can transfer via atmospheric dust fall or rainwater into the soil and endanger human health via food chains. Therefore, efficient utilization of CG has been an increasing concern and has drawn much attention in China. Many engineering studies on reusing CG have reported that coal gangue gave better properties to concrete [11–13], brick [14], and backfill [15–17]. However, most CG still accumulates on the Earth's surface. Heavy metals from coal gangue have influenced the natural environment, the ecological environment, and human health through leaching, the process of extracting metals from the coal gangue by dissolving them in variable pH liquids. Therefore, leaching has proven to be one of the main pathways for trace elements to enter the ecosystem [18–26].

Several heavy metals—As, Co, Cr, Cd, Cu, Mn, Ni, Se, Sn, V, Zn, and Pb—could lead to potential environmental impacts [3,27,28]. Trace element Cr was selected for investigation in this present study to assess their leaching behavior in CG and its relationship.

A few researchers have paid attention to the pH value in different locations and their relationship with heavy metals [29]. Some research projects have been conducted to elucidate methods for restraining heavy metal pollution by absorbing the heavy metals of CG under different water environments. Clay minerals montmorillonite, illite, and smectite are suitable for adsorption of heavy metals [30]. Because expansive soil (ES) contains a large amount of clay minerals, especially montmorillonite, this study focuses on the selection of expansive soil to control heavy metal to reduce their environmental pollution.

Covering CG with a layer of compacted soils can minimize risks related to the leaching [31–33]. A leaching test is an effective method to simulate the process of extracting substances from a solid by dissolving them in a liquid and assessing the potential impact on water quality. The metal concentrations dissolved out, and the release rate of heavy metals from coal gangue was related to the pH value of leachate [34]. The environmental geochemistry effect and change regularity of heavy metal in coal gangue in the process of weathering were studied, and it was found that pH was an important factor influencing the dissolution of heavy metals in the static soaking of coal gangue. A leaching test of the coal gangues was designed to study the migration mechanism and regularity of the harmful metals, and it was found that metal concentrations dissolved out, and the release rate from coal gangue was related to the pH of leachate and the element concentrations in coal gangues [34].

In view of the facts mentioned above, this study aimed to investigate the leaching behavior of Cr, one of the most toxic metals, from CG. To achieve the objectives of reuse of CG and efficiency in the removal of Cr, in this study, column leaching experiments were carried for removal of Cr using expansive soil as the modifier, and unconfined compressive strength (UCS) tests were conducted according to the different contents of expansive soil. The objectives of this study were to assess the strength of CG from Dongsheng and Wuhai, to determine a reasonable proportion of expansive soil in the mixture to retard heavy metal accumulation in groundwater, and to provide a scientific basis for recycling CG used as an embankment filler composed of expansive soil as its major composition besides the CG.

## 2. Materials and Methods

### 2.1. Sampling and Specimen Preparation

There were two types of materials used in this research. The first was an expansive soil from Gaomiaozi (located at 40.795519, 114.015291), Jining, Inner Mongolia, China, and the second was a coal gangue from sampling sites located at Yongshun Coal Mine (located at 39.864657, 110.031856), Dongsheng, and Nalingou Coal Mine (located at 39.310209, 106.883062), Wuhai, Inner Mongolia, China.

The chemical composition (see Table 1) [35] and the physical characteristics (see Table 2) of the experimental expansive soil passed through a 0.5 mm sieve were determined according to the test procedure [36].

**Table 1.** Chemical composition of expansive soil (wt %).

| Sample | $SiO_2$ | $Al_2O_3$ | $Fe_2O_3$ | CaO | MgO | Cr |
|--------|---------|-----------|-----------|-----|-----|-----|
| ES | 69.17 | 14.43 | 3.12 | 1.29 | 3.31 | 0.0008 |

**Table 2.** Physical characteristics of expansive soil.

| Parameter | Liquid Limit/% | Plastic Limit/% | Plasticity Index | Free Expansion Ratio/% |
|-----------|----------------|-----------------|------------------|------------------------|
| Values | 57.2 | 28.7 | 28.5 | 46 |

Fresh test samples from coal mining of Dongsheng (DS) and Wuhai (WH), Inner Mongolia, China, were packed in bags and taken to the laboratory. The chemical components of the CG samples were different depending on their produce sites. The mineral composition of two coal gangues obtained by X-ray Diffraction (XRD) and X-ray fluorescence spectroscopy (XRF) are shown in Figure 1. Table 3 summarizes the chemical composition (wt %) of these CG samples [37], and Table 4 shows their basic physical and mechanical indexes [38], determined according to the test methods [38,39].

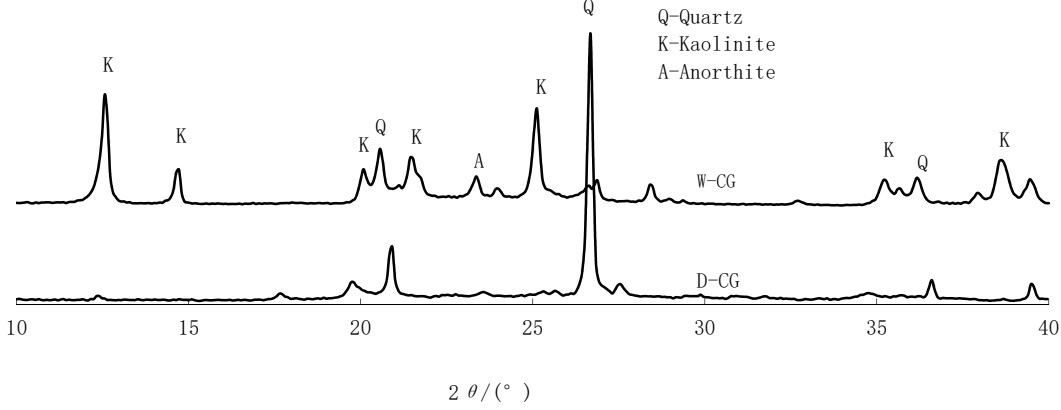

**Figure 1.** Mineral composition of coal gangues (CGs).

From Figure 1 and according to the literature [40], the spectrum of the sample shows a sharp peak, indicating that the mineral was a crystal structure. Major minerals include quartz, kaolinite, and anorthite. The main chemical components of two coal gangues were $SiO_2$ and $Al_2O_3$, while Wuhai coal gangue (WH-CG) contained more $SiO_2$ than Dongsheng coal gangue (DS-CG) (as shown in Table 3).

**Table 3.** Chemical components of CG (wt %).

| Sample | $SiO_2$ | $Al_2O_3$ | $Fe_2O_3$ | CaO | MgO | C | Cr |
|--------|---------|-----------|-----------|-----|-----|---|-----|
| DS-CG | 58.50 | 25.70 | 6.60 | 1.50 | 1.10 | 3.6 | 0.02 |
| WH-CG | 62.10 | 24.80 | 4.30 | 1.80 | 1.20 | 2.9 | 0.02 |

It is shown in Table 4 that Dongsheng-CG (DS-CG) has a stronger water absorption rate and higher crushing value than Wuhai-CG (WH-CG).

**Table 4.** Basic indexes of CG (%).

| Sample | Water Absorption Rate | Burn of Rate | Crushing Value |
|--------|-----------------------|--------------|----------------|
| DS-CG | 3.7 | 13.2 | 24.2 |
| WH-CG | 0.5 | 14.4 | 20.6 |

The natural CG gradation was poor due to the proportion of large grain particles, which meant that the CG could not directly be used as embankment material. The requirements of an embankment are a coefficient of uniformity (Cu) greater than 5, and a coefficient of curvature (Cc) between 1 and 3. Instead, the CG must be optimized through crushing to reduce particle size, as in [41], or by the addition of smaller particles followed by compaction. Initially, in this work, the CG was broken by a crusher and then was ground and sieved through a mesh with a sieve of 5 mm; finally, the CG was collected and stored in sterile plastic packages. The gradation curves are shown in Figure 2.

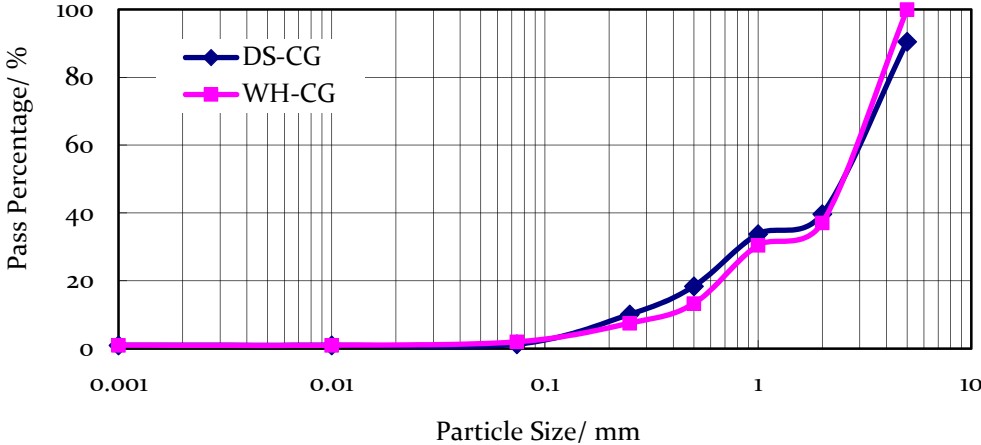

**Figure 2.** Gradation curves of CG samples.

Based on Figure 2, Cus for DS-CG and WH-CG were found to be 11.92 and 7.50, respectively, both of them greater than 5, while Ccs were 0.79 and 0.83, respectively, and neither were distributed between 1 and 3. The gradation of CG particles appeared to be defective, as it included a high percentage of coarse particles.

### 2.2. Experiments Methods

The background concentrations in $H_2O$ of the tested solutions of chemical reagents were 1000 mg/L. Concentrations were analyzed by digestion of dried samples in a mixture according to the standard of GSB 04-1723-2004 [42].

In this article, only the leached concentrations of Cr to characterize the leaching behavior were considered. The role of the factors affecting the leaching behavior, including leaching time and expansive soil addition amounts, were simultaneously investigated. Firstly, the mixtures were prepared by adding 0 wt %, 10 wt %, 20 wt %, 30 wt %, and 40 wt % of expansive soil into CG samples, which were then mixed. The test setup and the testing procedure are illustrated in Figure 3.

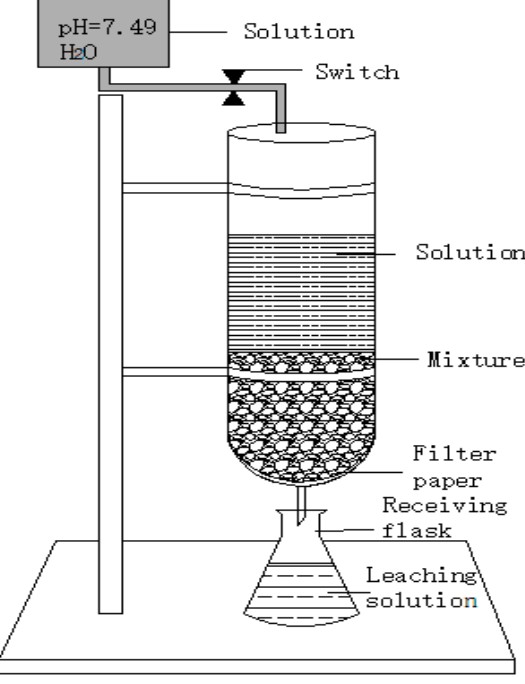

**Figure 3.** Column experiment setup.

Thereafter, 200 mL of drinking water (pH = 7.49) was poured one time every 24 h over five days. Leaching solution was collected daily and analyzed.

The pH values of the tested solutions were determined by a pH meter (Model PHS-3C, Shanghai INESA Scientific Instrument Co., Ltd., Shanghai, China). Cr was selected for investigation in this study to assess its leaching behavior in CG and the engineering properties of CG. The concentrations of Cr were measured with the atomic absorption spectrometer (Model 4510GF, Shanghai INESA Scientific Instrument Co., Ltd., Shanghai, China), which had a wavelength of 213–357.8 nm, a negative high pressure of 228–272 V, and a slit of 0.2 nm. Three groups of parallel tests were performed with a blank control for each sample to control the relative error within 5%. The final leached sample was detected for the unconfined compressive strength value of the material in an unconfined compressive strength meter (YSH-2, Nanjing Soil Instrument Factory Co., Ltd., Nanjing, China) in the lab to evaluate the engineering properties of coal gangue with expansive soil.

## 3. Results and Discussions

### 3.1. pH of Leaching Solutions

The pH values were determined at intervals of 24 h, and the obtained results are shown in Figure 4.

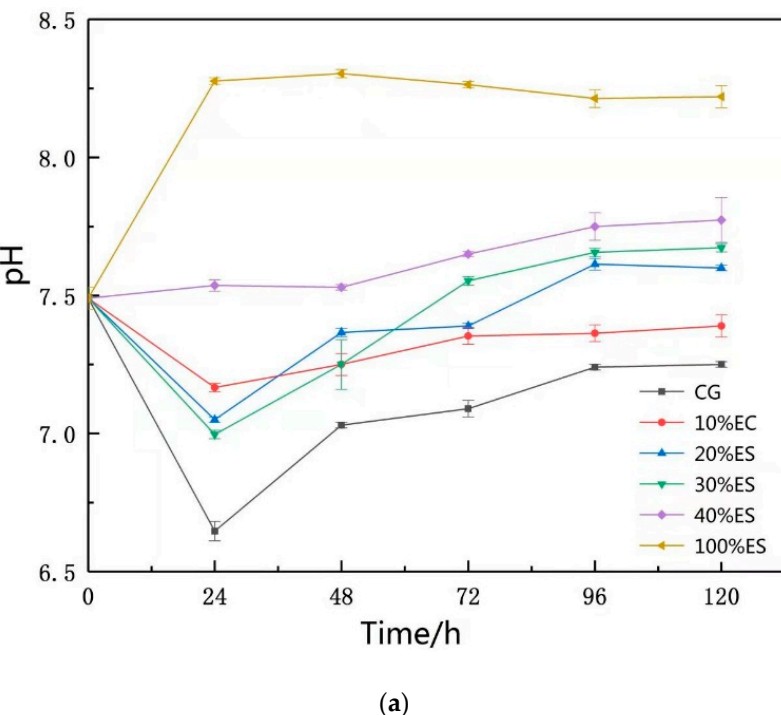

(**a**)

**Figure 4.** *Cont.*

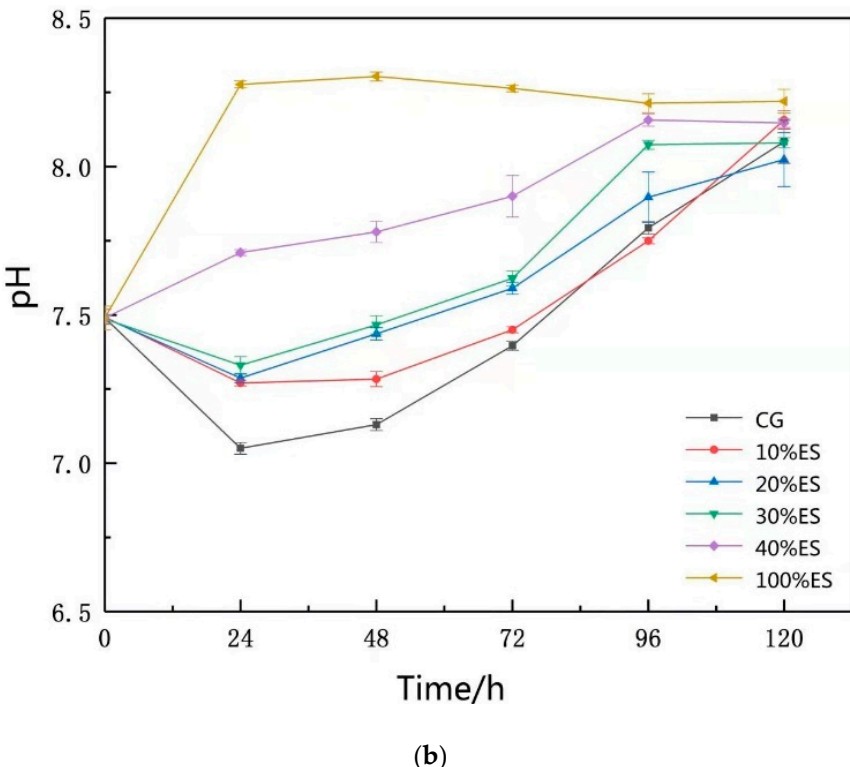

(**b**)

**Figure 4.** pH values of leaching solutions for CG. (**a**) DS-CG; (**b**) WH-CG.

The pH values of different expansive soil addition in the resulting leachates were mutative at different leaching periods (Figure 4). The pH value of 100% ES leaching solution peaked at 24 h, and then the pH value fluctuated at an alkaline state of about 8.25. From the leaching beginning, pH value was extremely reduced, reaching a minimum at 24 h, followed by gradual increase for the CG curve. Finally, the values stayed at around 7.25 and 8.10 at 120 h for DS-CG and WH-CG, respectively. The pH curves after the addition of expansive soil were higher than the CG curve, indicating that the pH value was enhanced by the influence of expansive soil incorporation. The curve of 40% ES was lower than that of 100% ES, and the pH value kept increasing and remained alkaline throughout the leaching process. However, pH values of CG with 10% ES, 20% ES, and 30% ES decreased to the minimum at 24 h because acidic substances ($SiO_2$) in CG were dissolved and then increased as a result of the effect of alkaline matters (MgO and CaO) in the samples. In DS-CG, WH-CG, and 100% ES, the total amounts of MgO and CaO were 2.6, 3.0, and 4.6 in Tables 1 and 3. A possible explanation is that alkaline substances dissolved and partly neutralized the previous acidic substances, thus the solution tended towards a neutral state and an alkaline state afterwards.

### 3.2. Concentration of Leaching Solutions

Different trend curves for the leached cumulative concentration of Cr were changed by leaching time and could be measured, and they are presented in Figure 5.

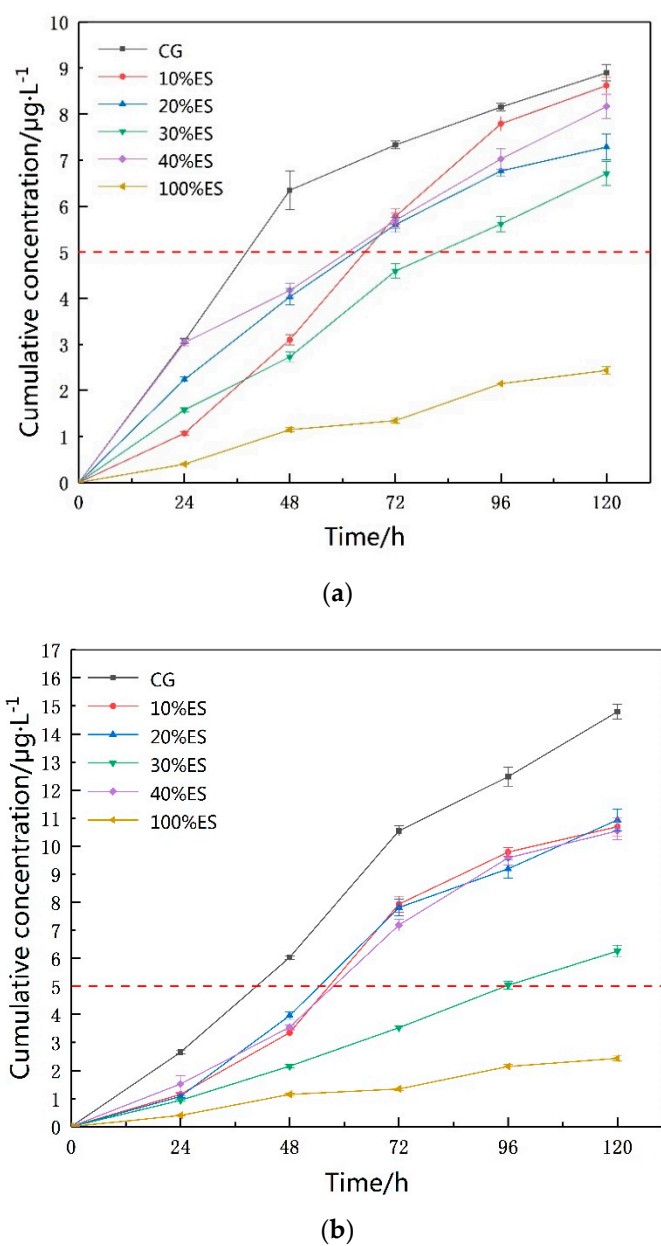

**Figure 5.** Leached cumulative concentration of Cr. (**a**) Concentration for DS-CG; (**b**) concentration for WH-CG.

Ascending curves were observed for the cumulative concentration of Cr with leaching time (Figure 5). The red line with ordinate 5 μg/L is the limitation of the quality standard for groundwater in China. The cumulative concentration curve of 100% ES showed lower and total concentration values of no more than 2.5 μg/L. The cumulative concentration of CG increased to the red line after experiencing leaching for about 1.5 days, while different curves with added expansive soil went through different leaching times and exceeded this limit line about 2.5 days later. In short, the cumulative concentration of Cr from CG was affected by leaching time and expansive soil addition amount.

### 3.3. Leaching Time

Leaching time is one of the important factors impacting the leaching behavior of trace elements from coal gangue [43,44]. The effect of leaching time on the leaching behavior of the trance elements was investigated in this study. The concentrations of elements in the resulting leachates reached maximum at different leaching times (Figure 5). From Figure 5a, the DS-CG curve shows that the

leaching time experienced by the cumulative concentration of leaching solution reaching the limit line was 36 h, which was faster than adding expansive soil. However, with 30% ES, until leaching for 80 h, the cumulative concentration reached the limit value. With 10% ES, 20% ES, and 40% ES, the cumulative concentration could go through up to 60 h and would not get to the limitation. Different from DS-CG, curves in Figure 5b attaining the limit required longer leaching time for WH-CG. Around 54 h were needed for CG with 10% ES, 20% ES, and 40% ES; however, 30% ES demanded 96 h. The reason for this could be that heavy metal elements were absorbed on the surface of the solid materials and formed a water solution, which could easily penetrate the soil [45]. As leaching progressed, the cumulative concentration increased gradually with leaching time owing to elements that were precipitated from internal solid materials, which needed to experience a longer time than surface elements.

### 3.4. Expansive Soil Amount

The amount of added expansive soil was also one of the factors impacting the leaching behavior of Cr in the leaching test. The examination of the results showed in Figure 5 revealed that, during the process of water leaching, the cumulative concentrations of Cr from DS-CG and WH-CG mixtures changed for the different expansive soil contents. Expansive soil of 30% reduced the cumulative concentration of DS-CG and WH-CG by about 25% and 60%, respectively, at leaching times of 120 h. Furthermore, incorporating expansive soil into coal gangue could extend the time to approach or reach the red line, thus ensuring the safety of the environment. The mixture with added expansive soil formed a system of the coarse CG particles surrounded by the fine particles of expansive soil. Therefore, leached concentrations of Cr were reduced by adding expansive soil when compared to the original CG leachate. The best combination to reduce Cr leaching from CG was found to be 30% ES.

## 4. Evaluation

### 4.1. Potential Ecological Risk

Chromium can exist in two states, namely trivalent [Cr (III)] and hexavalent [Cr (VI)] forms in an aqueous environment. Cr (VI) is highly soluble, while Cr (III) is relatively insoluble [46]. This study used a water leaching method. The chromium eluted from coal gangue was Cr (VI) with a high ability to be dissolved, and its hazard and mobility were greater than those of Cr (III). The cumulative concentration of Cr from CG in the leaching solution at 120 h in comparison with the limitation of the quality standard for groundwater [47] and the World Health Organization (WHO) [48] is listed in Table 5.

**Table 5.** Cumulative concentration of Cr and its limitations ($\mu$g/L).

| Leachate of DS-CG | Leachate of WH-CG | Limitation of the Quality Standard for Groundwater | Limitation of World Health Organization |
|:---:|:---:|:---:|:---:|
| 8.9 | 14.7 | ≤5 | ≤50 |

The cumulative concentration of Cr in the leaching solution was greater than the limitation of the quality standard for groundwater but less than the WHO standard. Considering that the requirements of the WHO are relatively broad, that there are strict demands of the national standard, and that Cr has a very serious impact on the human body and the environment, it was necessary to incorporate the adsorbent to control the Cr.

### 4.2. Unconfined Compressive Strength

Unconfined compressive strength is one of the important parameters in quality inspection and assessment of engineering. The relationship between the UCS and the expansive soil content is shown in Figure 6.

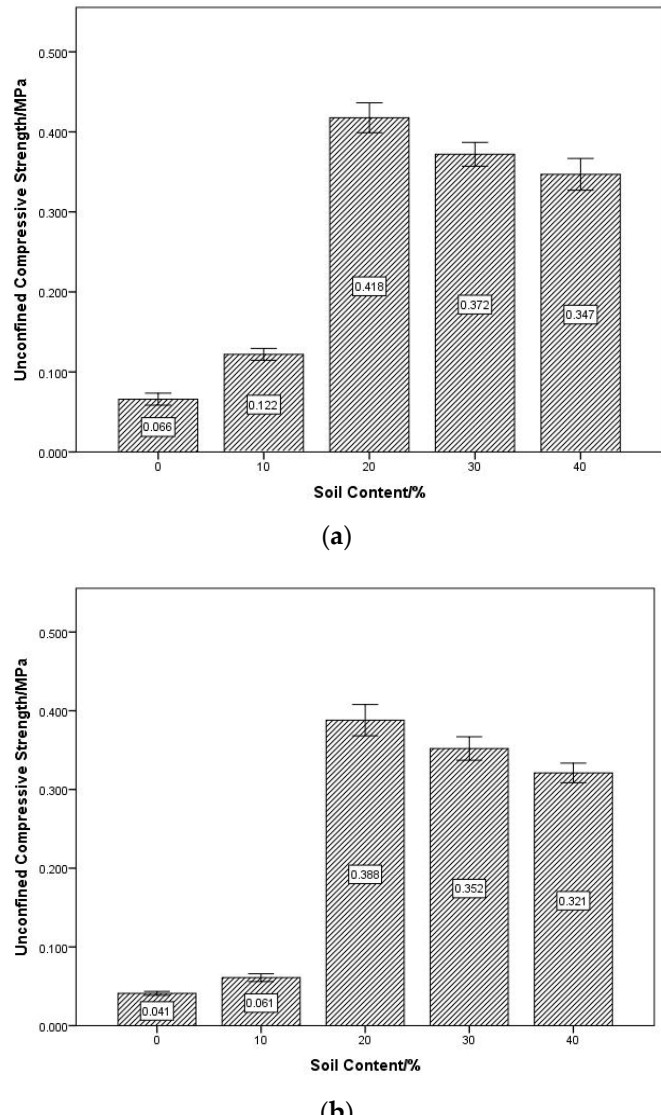

**Figure 6.** Unconfined compressive strength (UCS). (**a**) DS-CG; (**b**) WH-CG.

The examination of Figure 6 illustrates that the soil content had a great influence on the UCS of the specimen. The UCS of coal gangue mixed with expansive soil was higher than that without soil in both DS-CG and WH-CG. In terms of expansive soil content, the UCS of 10% expansive soil increased by around two times and one and a half times in DS-CG and WH-CG, respectively, while more than 10% expansive soil raised it at least four times in both mixtures and reached peaks of 0.38 MPa and 0.41 MPa, respectively, with 20% soil content.

## 5. Conclusions

Leaching concentrations of Cr from CG in the west of Inner Mongolia, China were investigated in this study. Based on the results of leaching behavior, the following conclusions were drawn from the experimental results and analysis:

(1) The two kinds of CG have a slight difference in chemical components; therefore, there were some differences in $H_2O$ after leaching. The pH values of different expansive soil additions in the resulting leachate were variable at different leaching stages. The pH values of leachate solutions for mixtures with expansive soil were higher than those with no expansive soil both in DS-CG and WH-CG. The pH value was enhanced by the influence of expansive soil incorporation and was alkaline, which could adsorb Cr leaching from coal gangue.

(2) During the process of water leaching, cumulative concentration of Cr in leachate from coal gangue was reduced with an increase in expansive soil. Adding expansive soil showed a binding ability on Cr when compared with original CG. Different additions of expansive soil had different effects on Cr from leaching solutions. In the view of leaching cumulative concentration, expansive soil of 30% was suitable to prohibit the dissolution of Cr.

(3) The coal gangue with 10% ES presented larger unconfined compressive strength than that of coal gangue. In particular, 20% ES, 30% ES, and 40% ES presented at least four times the UCS of coal gangue.

In conclusion, the modification of CG taking advantage of expansive soil as a modifier is a feasible industrial technique. Applying expansive soil can not only suppress Cr in coal gangue but can also improve the UCS of coal gangue. Therefore, for actual projects such as embankment filling and backfilling, the expansive soil can be mixed into coal gangue to form a mixture for road construction, landfill, and other projects. It could meet the engineering strength requirements as well as achieve waste recycling and reduce environmental pollution.

**Author Contributions:** Conceptualization, R.Z. and Y.Z.; experiments, R.Z.; data curation, R.Z.; writing—original draft preparation, Y.Z.; writing—review and editing, Y.Z. and H.B.

**Funding:** This research was funded by the National Natural Science Foundation of China, grant number 51669025 and China Scholarship Council.

**Conflicts of Interest:** The authors declare no conflict of interest.

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
