# Peer review of "Evaluation for the Leaching of Cr from Coal Gangue Using Expansive Soils"

_processes, doi:10.3390/pr7080478_

Reviewer 1 Report

Line 78. The coordinates of the coal gangue written incorrectly. Use this form: XX.XXXXXX, XX.XXXXXX.

Chapter 2.1 Please, add an XRD pattern of 2 types of coal gangue.

Table 1. Please add information about Cr content in coal gangue.

Figures 3-4. Use different colours to mark the curves. It will help the understanding of these figures.

Figure 4. What explains the peak of 10% ES for DS-CG?

Figure 4. Why do high chromium content peaks appear when using 20, 30, 40% at 48, 72 and 96 hours for WH-CG figure 4?

Chapter 3.1 What is the chemical content of Cr in water dangerous for humans? Are the contents of Cr represented in figures 3-4 dangerous? What are the standards for China and the World Health Organization? Do they correlate? What form of chromium is in water, what is its valence?

In all figures, you need to add errors bar.

Conclusion. How will the research results be used when applied in real life? Will coal gangue be mixed with the soil and stored in open areas?

Reviewer 2 Report

The paper “Evaluation of a Sustainable Solution for the Leaching Behavior Coal Gangue Using Expansive Soils” is poorly conducted and written, and has little to no value to the scientific community. No clear motivation for this specific study is given, the experiments are poorly designed and testing unsubstantiated. The results are unconnected and unintegrated.

Introduction:

The introduction does not adequately motivate the study and is poorly written. No explanation of the leaching process is given (reaction mechanism), thus no motivation for liberation of Cr is given nor reason for measurement variables (e.g. pH). Critically, a number of issues exist with the expressed aims of the study. No definition of expansive soil (the focus of the paper) is provided nor any explanation of its importance or potential impact. Why are UCS tests important (no explanation is given)? Measurement of pH and Cr concentration are not objectives – they are results. Furthermore, what are “engineering properties of CG”? This is a non-statement.

Materials and methods:

What are the composition and physical characteristics of the expansive soil? This needs to be given for the findings to be interpreted. No details of the requirements (size) of an embankment material is given, thus the statement in L89-90 is unsubstantiated. L99 inadequate for what? Critically no Cr content of the samples is given! What is its concentration and what is its chemical form?

L102 background concentration in what solutions? L104 reference inadequate. L108 no explanation of what the % indicates – weight, volume etc. L109 what does “mixed evenly” mean? L113 how is liquid collected by filter paper?

Results and Evaluation:

Claim of lowest pH after 24 hours is only true for 2 of the samples. 2 reach a minimum after 48h, after initially (24h) staying unchanged. No substantiation of “acidic substances” given – what are they? Subsequent increase was not steady. Overall, analyses of pH trends is problematic.

Concentration results are also problematic. For both samples, the CG results are lower than some of the mixed results which makes no sense. Overall, no trends are observable.

Leaching time and expansive soil amount sections do not say anything of significance.

Table 3 indicates that there was no risk with the original samples – so why do the study!?

Overall:

No clear conclusions can be made from the study. Key reasons for this are that no original Cr content is given for the samples, no error analysis is included (e.g. for the UCS tests to confirm the peak) nor any motivation given for the testing conditions and leaching mechanisms. Critically, the test are not able to explain if any changes in trends are due to (i) sample variation, (ii) change in leaching mechanism or reactions, (iii) change in fluid transport properties or (iv) simply dilution of the CG due to inclusion of the ES.

The quality of the writing and captions makes the paper very difficult to read and understand and needs a major edit.

Some initial comments are provided, but issues were so numerous that annotations are not provided for the whole document:

L38-39 Sentence does not make sense

L39-41 Sentence does not make sense

L41 “then main pathway”

L42 “elements to enter”

L43 Which trace elements?

L43 Why “therefore”? No motivation has been given.

L46 “and its relationship”

L49-51 does not make sense

L51 water does not have a “permeability”. A solid matrix has a permeability which affects the ability of water to flow through it.

L53 no explanation of what “expansive soil” is given, nor any motivation of why it should have been studied

L55 “A leaching test”

L55 what process?

L55 impact of what?

L55-57 Sentence does not make sense. How can “metal dissolve out of heavy metal”?

Reviewer 3 Report

Comments to the manuscript “Evaluation of a sustainable solution for the leaching behavior coal gangue using expansive soils”

Manuscript is related to important problems – utilization of coal mining wastes and lowering of heavy metals pollution of mining areas. Results are interesting and worthy to be published but several corrections are needed. Careful  linguistic verification by native speaker of the manuscript is needed before preparation of the final version  after corrections.

1.Title is too general. Leaching of Cr is discussed. “Leaching behaviour of coal gangue” suggests that chemical composition of leachates will be discussed. I suggest modification of the title.

2.Coal gangue is described as “main by-product’ (line 28). Is it a by-product of coal production according to legal regulations or it is a waste?. In line 30 we can find “… coal gangue are stacked to form huge solid waste dumps…”.

3.Line 44: “Trace elements, Zn, Ni, Cr, Cd, and Pb, were therefore selected for investigation in the present study…”. Presented results are limited to leaching of Cr. Introduction should be corrected to correspond with the presented results.

4.Line 64: “…. This study was aimed to investigate the leaching of Cr …”. This information contradicts to this from line 44.

5.Chemical composition of coal gangue (Table 1, line 83). Data are presented probably as wt% (please add this information). Method of chemical analysis is not mentioned. “C” in Table 1 means “Total carbon content”? If it possible to supply data related to heave metals content, please do it. Content of Cr is needed for discussion of leaching of Cr.

6.“Basin indexes of GC” (line 84). Methods of determination of these parameters are needed.

7.Line 97: Coefficient of uniformity and coefficient of curvature are not defined in the manuscript. “From Figure 1 the coefficient of uniformity (Cu) ….. were found to 40.5 and 10.19 …. While the coefficient of curvature (Cc) are 5.25 and 1.16…”. Values are not marked in Figure 1.

8.If possible to add data on mineral composition of coal gangue, please do it. Mineral composition is important in discussion of leaching properties of elements. Is it possible to describe form of occurrence of Cr bearing components in coal gangue?

9.Line 103: “Heavy metal concentrations” means “Cr concentrations”? Content of Cr in leachate doesn’t correspond to concentration in coal gangue.

10.Line 104: “Standard of GSB 04-1723-2004” is not described or cited in references section.

11.Line 112: Is it proper result of measurement of pH of deionized water? pH=7.49?

12.Line 115: “Cr (DS-Cr and WH-Cr) were selected …” . Perhaps “was selected”?

13.Line 124: Is it possible to present results pH determination for 100% ES? It will help to understand variations of pH in different samples with time.

14.Line 134: Is it possible to observe “a rise (of pH value) at 24 to 48 hours”? Irregular variations can be observed.

15.Discussion related pH variation is needed. In the manuscript we can find precise description but not an explanation (or attempt of explanation).

16.Line 146: Is it possible to add data related to leaching of Cr from ES? What is the content of Cr in ES?

17.Line 182: “The best combination to reduce Cr leaching from CG was found to be 120 h and 20% of Expansive soil”. The statement results from Figure 4. It is possible that during prolonged leaching soluble components containing Cr were dissolved. Longer leaching time is needed to observe if a “plateau” of Cr concentration was obtained.

18.Total content of Cr in coal gangue is necessary to discuss what means reduction of leaching of Cr? Dissolution of Cr containing components?

19.Line 186: Environmental Quality Standard is not cited in references. Perhaps Cr instead of Ni?

20.Line 188: What is test value presented in Table 3. Where is the source of these data?

21.Line 212: “… Cr in DS-CG was reduced …” or “… Cr in leachate from DS-CG… “?

22.Line 215: What means: DS-Cr and WH-Cr?

Author Response

Round  2

Reviewer 1 Report

Figure 1. Need to write the full name of the minerals. 

Line 90-91. It is necessary to add mineral names like quartz (SiO2) and etc.

Line 204-205. Whats the chromium valency in your work. This is very important information since valence (VI) is extremely toxic to humans.

Reviewer 2 Report

We thank the authors for incorporating the suggested changes to the paper. It has been done to a sufficient level. There is still opportunity to improve the study and presentation, but motivation and content are now more substantiated and relevant.

There are still multiple grammatical and spelling errors throughout the document. This needs to be corrected before publication.

Author Response

Dear reviewer,I am very grateful your suggestion, the amendment as below.

Suggestion: We thank the authors for incorporating the suggested changes to the paper. It has been done to a sufficient level. There is still opportunity to improve the study and presentation, but motivation and content are now more substantiated and relevant.

There are still multiple grammatical and spelling errors throughout the document. This needs to be corrected before publication.

Revision: checked and modified.